# Improving Pancreatic Cyst Management: Artificial Intelligence-Powered Prediction of Advanced Neoplasms through Endoscopic Ultrasound-Guided Confocal Endomicroscopy

**DOI:** 10.3390/biomimetics8060496

**Published:** 2023-10-19

**Authors:** Joanna Jiang, Wei-Lun Chao, Troy Cao, Stacey Culp, Bertrand Napoléon, Samer El-Dika, Jorge D. Machicado, Rahul Pannala, Shaffer Mok, Anjuli K. Luthra, Venkata S. Akshintala, Thiruvengadam Muniraj, Somashekar G. Krishna

**Affiliations:** 1Division of Gastroenterology, Hepatology, and Nutrition, Department of Internal Medicine, The Ohio State University Wexner Medical Center, Columbus, OH 43210, USA; 2Department of Computer Science and Engineering, College of Engineering, The Ohio State University, Columbus, OH 43210, USA; 3College of Medicine, The Ohio State University, Columbus, OH 43210, USA; 4Department of Biomedical Informatics, College of Medicine, The Ohio State University, Columbus, OH 43210, USA; 5Department of Gastroenterology, Jean Mermoz Private Hospital, 69008 Lyon, France; 6Division of Gastroenterology and Hepatology, Stanford University, Stanford, CA 94305, USA; 7Division of Gastroenterology, University of Michigan, Ann Arbor, MI 48109, USA; 8Division of Gastroenterology and Hepatology, Mayo Clinic Arizona, Phoenix, AZ 85054, USA; 9Division of Gastrointestinal Oncology, Moffitt Cancer Center, Tampa, FL 33612, USA; 10Division of Gastroenterology, Johns Hopkins Medical Institutions, Baltimore, MD 21287, USA; 11Department of Internal Medicine, Yale University School of Medicine, New Haven, CT 06510, USA

**Keywords:** pancreatic cancer, pancreatic cysts, IPMN, artificial intelligence, machine learning, endoscopy, endoscopic ultrasound, EUS-nCLE, endomicroscopy

## Abstract

Despite the increasing rate of detection of incidental pancreatic cystic lesions (PCLs), current standard-of-care methods for their diagnosis and risk stratification remain inadequate. Intraductal papillary mucinous neoplasms (IPMNs) are the most prevalent PCLs. The existing modalities, including endoscopic ultrasound and cyst fluid analysis, only achieve accuracy rates of 65–75% in identifying carcinoma or high-grade dysplasia in IPMNs. Furthermore, surgical resection of PCLs reveals that up to half exhibit only low-grade dysplastic changes or benign neoplasms. To reduce unnecessary and high-risk pancreatic surgeries, more precise diagnostic techniques are necessary. A promising approach involves integrating existing data, such as clinical features, cyst morphology, and data from cyst fluid analysis, with confocal endomicroscopy and radiomics to enhance the prediction of advanced neoplasms in PCLs. Artificial intelligence and machine learning modalities can play a crucial role in achieving this goal. In this review, we explore current and future techniques to leverage these advanced technologies to improve diagnostic accuracy in the context of PCLs.

## 1. Introduction

The widespread use of cross-sectional imaging, such as computed tomography (CT) and magnetic resonance imaging (MRI), has resulted in a high incidence of incidentally detected pancreatic cystic lesions (PCLs). There is currently an “epidemic” of such lesions, with 15–45% of asymptomatic patients having a pancreatic cyst identified in cross-sectional abdominal imaging studies [1]. PCLs encompass a broad range of lesions, ranging from benign cysts to mucinous pre-malignant lesions that carry a risk of progressing to high-grade dysplasia or adenocarcinoma (HGD-Ca). Surgical interventions, such as pancreaticoduodenectomy (Whipple’s procedure), total pancreatectomy, and distal pancreatectomy, aim to resect malignant lesions with HGD-Ca. Conversely, patients with only low-grade dysplasia (LGD) can be managed through serial imaging monitoring [2,3,4].

PCLs are classified into mucinous and non-mucinous lesions. Non-mucinous lesions include cystic neuroendocrine tumors, solid pseudopapillary neoplasms, and serous cystadenomas. Mucinous cysts include intraductal papillary mucinous neoplasms (IPMNs) and mucinous cystic neoplasms (MCNs), both of which are pancreatic cancer precursors. IPMNs are classified as main-duct (MD) IPMNs, which represent cystic dilation of the main pancreatic duct, and branch-duct (BD) IPMNs, which are cysts that lie in communication with the main duct. BD-IPMNs are the most common PCLs, with the reported risk of malignancy ranging between 6 and 46% [5,6,7]. The current standard of care for risk stratifying PCLs involves the use of endoscopic ultrasound (EUS)-guided fine needle aspiration (FNA) and analysis of cyst fluid, including measurements of carcinoembryonic antigen (CEA), cytology, amylase, and glucose levels. However, these modalities provide only 65–75% accuracy in identifying high-grade dysplasia or adenocarcinoma (HGD-Ca) [8]. 

It is estimated that two-thirds of surgically resected PCLs demonstrate only LGD or benign neoplasms, indicating a significant rate of overtreatment and unnecessary surgeries [9]. Considering the high morbidity and mortality of Whipple procedures and pancreatectomies, these data point to an unacceptably high false-positive rate with current diagnostic modalities [10,11]. Conversely, several series report that up to 37% of invasive cancers are discovered during routine follow-ups of suspected BD-IPMNs, suggesting possible delays in diagnosis [12]. Application of the Fukuoka International Consensus Guidelines specifically intended for IPMNs also continues to contribute to missed cancers at one end and surgical overtreatment and unnecessary pancreatic resections at the other [4,8,13,14,15,16,17].

Unlike solid tumors, where tissue biopsy often guides diagnosis, there are currently no standard-of-care, consistently reliable options for obtaining tissue from the PCL epithelium prior to resection. Additionally, IPMNs can demonstrate a wide range of histologic features within a single cyst that vary from low-grade dysplasia (LGD) to invasive cancer, suggesting that intracystic micro-biopsies may not precisely sample the area with the highest degree of malignant progression [5,6]. This creates a need for more precise diagnostic techniques that use existing information, such as cross-sectional imaging, cyst fluid (CEA, glucose, cytology) analysis, next-generation sequencing (NGS) of cyst fluid, and novel imaging biomarkers (confocal endomicroscopy). EUS-guided needle-based confocal laser endomicroscopy (EUS-nCLE) features of IPMNs are highlighted in Figure 1.

### 1.1. Artificial Intelligence

Artificial intelligence (AI) is a technology that aims to mimic human intelligence to perform tasks, such as object recognition and decision making. AI encompasses several branches, among which machine learning (ML) approaches have attracted the most attention in recent years. 

ML constructs models that could perform tasks by learning from data. For instance, to differentiate dog images from cat images, ML requires a dataset of cat and dog images, each of which is manually labeled with the ground-truth category so that it can train the model (i.e., a neural network classifier) to correctly classify images.

Many kinds of ML models and algorithms have been developed. Logistic regression linearly combines the features in a data sample [18]. Decision trees build tree-structured decision rules to classify a data sample; random forests build multiple trees and output their joint decisions (e.g., majority voting) to make the classification more stable [19,20]. Support vector machines (SVMs) could construct complicated decision rules by applying the kernel tricks, which are a set of mathematical techniques used to transform the input data into a higher-dimensional feature space. Kernel tricks enable SVMs to classify non-linearly separable data by implicitly computing dot products between transformed feature vectors without explicitly calculating the higher-dimensional feature space [21,22]. Most of these algorithms require informative features (often hand-crafted) to be extracted from data samples.

Recently, deep learning approaches that aim to learn neural network (NN)-based models have shown unprecedented results in many application domains such as image recognition [23,24]. Compared to the aforementioned methods, NN-based models can often learn to perform feature extraction and decision (e.g., classification) in an end-to-end fashion. In image recognition and computer vision, convolutional neural networks (CNNs), which consist of layers of convolution filters, are widely applied to capture image patterns [25]. Several representative CNN models include AlexNet, VGGNet, and ResNet for image classification [26,27,28]; Faster R-CNN and YOLO for object detection (i.e., localize objects in an image with bounding boxes and classify them) [29,30]; U-Nets and DeepLab for semantic segmentation (i.e., classify each image pixel into a semantic category) [31,32]; and Mask R-CNN for instance segmentation, which localizes and segments objects using object masks [33].

### 1.2. Challenges in AI Application to Pancreas Imaging

There are multiple considerations when designing an AI algorithm to risk stratify PCLs, which include steps from identifying the pancreas in a CT or MRI image, to selecting imaging features that can predict the malignant progression of a cyst. The interpretation of pancreatic lesions presents a unique challenge for AI. First, the pancreas occupies a relatively small area (approximately 1.3%) in cross-sectional images [34]. It is also both irregularly shaped and highly variable in its location relative to other organs [35]. Pancreatic lesions can also often have similar radiographic features to the surrounding tissue.

Previous models have required significant pre-processing of images prior to AI interpretation. Wei et al. developed an SVM system to diagnose serous cystadenomas. This system utilized CT radiomics features combined with regions of interest that were marked by a radiologist [36]. Similarly, Chakraborty et al. employed manually segmented pancreas images, with manual outlining of the pancreas head, body, and tail, to train AI models for predicting high-risk IPMNs [37]. While such models may achieve high accuracies, the need for human pre-processing of images ultimately detracts from the goal of leveraging AI to reduce specialist workload (Table 1).

There has been a recent shift toward more end-to-end, independent, or self-supervised predictive algorithms for the interpretation of CT images. Javed et al. presented the first framework for automated 3D pancreas segmentation in CT images, enabling the identification and delineation of the pancreas without manual intervention [41]. In a similar vein, Lim et al. validated a CNN model for automating pancreas segmentation, achieving a mean precision of 0.87 in an internal cohort and 0.78 for external validation [42]. Si et al. developed a deep learning model for diagnosing pancreatic tumors, including IPMNs and pancreatic ductal adenocarcinoma (PDAC), based on unedited, non-annotated abdominal CT images [34]. They used images from 347 patients and reported an AUC of 0.871 and accuracy of 82.7% for all tumor types. Remarkably, the model achieved 100% accuracy for IPMN diagnosis, with surgical histopathology as the reference standard. A manual review of the images required approximately 8 minutes per patient, whereas the model only needed an average of 18.6 seconds, demonstrating that a fully end-to-end predictive algorithm can significantly reduce specialist workload and healthcare utilization.

## 2. Materials and Methods

In this review, we highlight existing applications for the AI-powered diagnosis of PCLs and describe an ongoing study performed by our group to improve their accuracy and generalizability. 

A literature search was performed in February 2023 via PubMed and Embase for publications within the last 20 years (2003–2023). Articles were included if they constituted primary literature describing original AI algorithms for diagnosing or predicting malignant potential of pancreatic cystic lesions. The search was conducted by three authors (J.J., T.C., S.J.K.). Search terms included synonyms of “IPMNs”, “PCLs”, “artificial intelligence”, and “machine learning”. Review articles and clinical case reports were excluded.

The primary objective of this review is to outline an ongoing multicenter, prospective initiative that details the study methodology for enhancing and prospectively evaluating a CNN-AI algorithm based on nCLE to improve presurgical risk stratification for BD-IPMNs. Furthermore, we suggest and assess an integrated diagnostic approach, which incorporates nCLE, cyst fluid analysis, and standard-of-care variables, with the aim of enhancing the accuracy of IPMN risk stratification.

## 3. Results

### 3.1. Utility and Accuracy of EUS-nCLE

EUS-nCLE provides real-time microscopic analysis of PCL epithelium without the need for high-risk endoscopic biopsy or surgical excision [43,44]. The PCL is visualized using EUS, and intravenous fluorescein is injected 2–3 minutes prior to imaging. A 19-gauge FNA needle preloaded with an nCLE miniprobe is advanced into the cyst until it opposes the cyst epithelium. The miniprobe is moved throughout the cyst cavity to assess different areas of its internal lining for approximately 6 minutes. After the video sequence is acquired, the miniprobe is removed, and the cyst fluid is aspirated for further analysis [45].

In our previous studies, we demonstrated that quantitative analysis of PCL epithelium using EUS-nCLE outperformed the current standard of care in diagnosing HGD-Ca and LGD in IPMNs [46,47,48]. These findings highlight the potential of EUS-nCLE to enhance diagnostic capabilities in this context.

We discovered several features that can be visualized on EUS-nCLE, which correspond to a higher histologic grade. For example, papillary epithelial width, as measured on nCLE images, suggests cellular atypia, while increased epithelial darkness on images is associated with nuclear stratification. These characteristics demonstrated high sensitivity in predicting HGD-Ca in an analysis of 26 BD-IPMNs. Papillary epithelial “width” and “darkness” exhibited sensitivities of 90% and 91%, respectively [48]. Surgical histopathology served as the reference standard in this analysis. Additionally, there was substantial interobserver agreement (IOA) among external nCLE experts in detecting HGD-Ca, with a κ value of 0.61 for epithelial width and 0.55 for darkness [48].

### 3.2. CNN-AI Algorithm for nCLE Analysis

By employing accurate AI-driven image analysis and recognition, the need for time-consuming manual quantification of papillary epithelial parameters by endomicroscopy specialists can be bypassed. In a recent post hoc analysis of video frames from the INDEX study, which included patients with histopathologically proven IPMNs, two CNN-based algorithms were developed to detect HGD-Ca in the lesions [45]. The first algorithm utilizes an instance-segmentation-based model, specifically Mask R-CNN, to detect and segment papillary structures. It then measures papillary epithelial thickness and darkness and employs these features for diagnosing HGD-Ca. This model achieved an accuracy of 82.9%. The second algorithm applies a CNN model, namely VGGNet, to directly extract features from the holistic nCLE video frames for risk stratification. This approach yielded an accuracy of 85.7%. In comparison, the accuracy of current society guidelines (AGA and Fukuoka) reached only 68.6% and 74.3%, respectively. These findings highlight the potential of AI-based approaches in improving diagnostic accuracy and outperforming existing guidelines in the assessment of IPMNs.

### 3.3. Improving and Prospectively Evaluating the Single-Center Algorithm

In addition to high accuracy, our single-center algorithm must also demonstrate the ability to incorporate new patient data. Many patients with PCLs require EUS with or without nCLE, with big data being stored as large video files. The mean duration of the unedited EUS-nCLE videos in our previous studies was approximately seven minutes, and the videos were manually shortened to under three minutes of high-yield portions [45,48]. Both algorithms in our pilot study used pre-edited videos where frames with artifact, blurring, or redundancy were removed by an nCLE expert. An algorithm that can be directly applied to unedited videos, without the need for manual editing, will greatly increase model efficiency, applicability, and generalizability.

Our future plans involve the development of a video summarization AI algorithm that will convert unedited nCLE videos into shorter, high-yield video clips, which will, in turn, improve the performance of our CNN-based algorithm [49,50,51]. This approach aims to streamline the diagnostic process for HGD-Ca detection in three steps. First, a CNN-based classifier will be used to classify edited video frames into “high-risk” and “low-risk” images based on the presence or absence of papillary structures (as an indicator of potential dysplasia) [52]. Second, an instance-segmentation-based algorithm will be employed to segment papillae and measure papillary epithelial thickness and darkness, enabling the grading of dysplasia. Concurrently, a holistic-based algorithm will utilize a CNN-based model to extract features from nCLE images, focusing on identifying HGD-Ca. The outputs of these two algorithms will be appropriately fused to leverage their complementary abilities and reduce uncertainty in the diagnostic process. Finally, image-level predictions from the entire edited video will be aggregated using a majority voting approach to diagnose HGD-Ca in each subject. By implementing these steps and integrating the video summarization algorithm, we aim to enhance the accuracy and efficiency of our diagnostic approach for HGD-Ca detection in PCLs.

### 3.4. Creating an Integrative Predictive Algorithm

While EUS-nCLE provides valuable information for PCL diagnosis, it represents only one source of patient data. Multiple clinical, demographic, genomic, and radiographic data points have been identified as significant predictors of HGD-Ca; incorporating all of these data into an integrative predictive algorithm will likely significantly improve accuracy. It is important to acknowledge that existing predictive models and expert consensus-led guidelines may oversimplify the impact of each variable considered. In routine medical practice, clinicians take into account patient demographics, apply guidelines such as Fukuoka-ICG, and often engage in multidisciplinary team discussions to assess the risk of malignancy associated with PCLs [13]. By leveraging an ML-powered integrative framework, we can potentially optimize diagnostic accuracy by incorporating and analyzing the available data in a more comprehensive manner. This approach aims to combine the expertise of clinicians with the analytical power of ML algorithms to improve the accuracy of diagnosing PCLs and risk stratifying them accordingly.

Molecular analysis using NGS so far has been shown to more reliably predict HGD-Ca in IPMNs as compared to the standard of care. In a 2015 composite analysis, the combination of genetic mutations in SMAD4, RNF43, TP53, and aneuploidy predicted HGD-Ca in IPMNs with a sensitivity of 75% and a specificity of 92% [52]. Another study showed that the presence of KRAS/GNAS along with additional mutations in TP53, PIK3CA, and PTEN produced 88% sensitivity and 97% specificity for BD-IPMNs with HGD-Ca [53]. 

Standard-of-care variables include clinical characteristics (age, gender, onset of diabetes, family history symptoms, pancreatitis history), serum CA 19-9, cyst fluid analysis (glucose, CEA, cytology) as well as cyst and pancreas morphology, as detected via CT/MRI/EUS (factors, such as size, wall, thickness, mural nodules, growth rate, and pancreatic duct diameter). We intend to integrate four sources: cyst fluid NGS, standard-of-care variables in expert consensus guidelines (Fukuoka-ICG 2017 version), manual/human EUS-nCLE results, and CNN-AI results [13].

The first component of our vision involves creating a logistic regression model that integrates the four elements mentioned to predict HGD-Ca or LGD in BD-IPMNs. The contribution of each variable will be assessed, and our sample size is expected to include at least 300 BD-IPMNs, with approximately 50% being HGD-Ca cases.

The second component aims to develop an ML-based approach for integrating the four data sources. Decision trees are chosen as the ML algorithm of choice due to their ability to handle both continuous and categorical variables seamlessly. Additionally, decision trees provide human-understandable explanations of the decision-making process, enabling experts to verify and interpret the learned decision trees. This feature facilitates the integration of the model into future management guidelines. Suitable options for doing this include using random forests and XGBoost, both of which are well-known ensemble methods over decisions trees that offer easy ways to control the model complexity to overcome over/under-fitting by controlling the number of input variables, tree depths/widths, and number of trees [54,55,56]. These enhanced models have the potential to diagnose HGD-Ca in BD-IPMNs with optimized diagnostic accuracy. Furthermore, by identifying the most significant contributors from the data sources, these models can potentially inform new clinical practice guidelines and improve the risk stratification of BD-IPMNs. 

### 3.5. Prior Integrative Algorithms

Integration of multiple data sources into AI algorithms to solve clinical problems related to PCLs and pancreatic malignancy have shown promising results, as illustrated in Table 2 [52,57,58,59]. In 2015, Springer et al. developed an algorithm that analyzed multi-parametric features (known as Multivariate Organization of Combinatorial Alterations or MOCA) to identify PCLs requiring resection [52]. The researchers combined composite clinical markers (such as age, presence of abdominal symptoms, and specific imaging results) and composite molecular markers (including aneuploidy and various gene mutations) in their MOCA algorithm. When evaluating the markers individually, the composite clinical marker and the composite molecular marker showed sensitivities of 75% and 77%, respectively, in identifying cysts that required resection. However, when used together, their sensitivity increased to 89%. Furthermore, the combination of molecular and clinical markers in the study resulted in a sensitivity of 94% for detecting IPMNs, an increase from the 76% sensitivity observed with the composite molecular marker alone. Subsequently, the same group developed CompCyst in 2019, which is a test built on the MOCA algorithm. This test categorizes patients with PCLs into three management groups (surgery, surveillance, or discharge) based on the evaluation of various clinical and molecular markers [58]. Overall, these advancements in combining molecular and clinical markers, as demonstrated by the MOCA algorithm and CompCyst, offer improved stratification and management of patients with PCLs. The test integrated three data elements: presence of VHL mutation but absence of GNAS mutation, decreased expression of a VEGF-A protein, and a combination of factors (solid component of cyst seen on imaging, aneuploidy, and presence of mutations in certain genes). Clinical, imaging, and molecular data were integrated in their test, and they produced higher accuracy compared to conventional clinical and imaging criteria. The authors estimated that widespread use of CompCyst may reduce the number of unnecessary surgical resections by 60%.

Other recent studies have also leveraged ML to perform integrative analyses to manage other malignancies. One study used a deep-learning-based stacked ensemble model to predict the prognosis of breast cancer from available multi-modal cancer datasets, including genetic data (gene expression, copy number variation) and clinical data (age, subjective timing of menstruation and menopause, timing of pregnancy and others) [64]. Integrative approaches have also been applied to determine prognosis of clear cell renal cell carcinoma and lung adenocarcinoma [65,66]. Additionally, it is well established in the literature that deep learning methods utilizing multiple modalities of input data sources (multimodal) outperform methods with a single source of input data (unimodal) [67,68,69]. Given the promising evidence regarding the success of previous integrative AI algorithms in addressing both clinical and non-clinical challenges, we anticipate that our next step, involving an integrative approach with an ML-powered model, will contribute to optimizing diagnostic accuracy and further enhancing the ability to predict HGD-Ca in branch-duct IPMNs.

## 4. Discussion

In this review, we present some existing applications of AI in diagnosing and predicting malignant potential of PCLs. We propose a comprehensive plan for designing an accurate, integrative, and scalable predictive algorithm for predicting the progression of branch-duct IPMNs using a combination of EUS-nCLE findings, clinical data, fluid analysis, and radiographic imaging. To reduce the workload of the time-consuming pre-processing of videos by specialists, we will develop a video summarization algorithm capable of automating the editing of nCLE videos. This algorithm will use AI to identify high-yield video clips that contain relevant information for analysis. Next, we will employ two models to risk stratify the IPMNs based on the selected video clips. These models will utilize advanced ML techniques to assess and predict the likelihood of neoplasia progression. Subsequently, we will develop an integrative algorithm that incorporates standard-of-care variables, nCLE videos, and NGS in cyst fluid to further enhance the accuracy of risk stratification. By combining multiple data sources and leveraging the power of AI, we aim to improve the predictive capabilities of our algorithm. By implementing these steps, our goal is to develop an advanced AI-powered algorithm that can accurately predict the malignant progression of IPMNs, providing clinicians with valuable insights for personalized patient management and decision making.

There are several foreseeable limitations to this model. Firstly, EUS-nCLE is typically only available at large academic institutions and referral centers. Patients who undergo these procedures usually reside in metropolitan regions or possess the resources to seek treatment at such institutions. This is in contrast to non-invasive imaging modalities, like CT and MRI or even routine cyst fluid analysis, which are more widely accessible. Consequently, the preference for using data from patients who have undergone EUS-nCLE may introduce a selection bias into the final algorithm.

Another source of selection bias arises from the necessity for surgical histopathology to establish a formal diagnosis. As a result, the data used for training the models must come from patients who have already been identified as high risk and have undergone surgical resection.

Additionally, the costs and resources required for implementing the AI algorithm, including software development, maintenance, and staff training, could pose barriers to widespread adoption, especially in lower-resource regions. As is the case with all applications of AI in medicine, our algorithm should be viewed as an adjunct to assist in decision making rather than as a replacement for clinical expertise. Lastly, our plan to enhance generalizability by utilizing data from multiple hospitals across the U.S. presents unique challenges related to patient privacy.

### Multi-Center Collaboration

To ensure the generalizability of our algorithm to a larger patient population, it is crucial to validate it in a multi-center cohort. Our aim is to include patients from at least ten tertiary care centers, and the development and validation of the multi-center algorithm will be carried out concurrently with the design of our improved CNN and integrative models. Combining data from multiple centers presents several challenges, including increased variability, domain discrepancies, and differences in devices and favored imaging modalities. Furthermore, the importance of data privacy, protection, and ownership has become increasingly recognized in recent years.

Even when transmitting anonymous data between centers, it is widely acknowledged that simply removing a patient’s name or medical record number is insufficient to protect their privacy. Ideally, collaboration and algorithm updates should be achieved without the need to share patient data across centers. This would require implementing privacy-preserving techniques and secure protocols that allow for the exchange of knowledge and model updates while maintaining data confidentiality. Addressing these challenges will be essential in order to ensure effective collaboration and advancements in algorithm development without compromising patient privacy.

To address these challenges, we intend to utilize federated learning (FL) and domain adaptation (DA) techniques. Traditionally, data used in ML algorithms are collected from various sources and processed in a centralized manner at a single site. ML models are then trained using these centralized data. In contrast, FL aims to train AI models while keeping the data decentralized at multiple data collection sites [70,71]. In the mainstream FL framework, a centralized server coordinates the model training process. Initially, the server broadcasts the ML model to each data collection site, known as a client. These client devices, which are secure servers located in their respective hospitals, then train the models using their local data. Subsequently, the updated model parameters are sent back to the server from each client. The server aggregates these client model parameters into a single ML model. This approach enables collaborative model development and adaptation among multiple hospitals without the need to share any patient data [71,72,73,74,75,76,77]. FL has demonstrated reliable results when applied to clinical information, PET images, and multimodal data. Building upon our prior work, we will develop an FL framework that leverages unlabeled data at our center to enhance the aggregation of clients’ models. This approach will further improve the performance and adaptability of the AI model while maintaining data privacy and security [78,79,80].

In addition to FL, we will also employ DA techniques to address the data domain discrepancy across different centers. It is widely recognized in the ML community that such domain discrepancy can negatively impact the accuracy of ML models. Traditional DA methods typically involve sharing data across different sources or centers to quantify the differences in data distributions and then applying transformations to reduce the domain discrepancy [81,82,83]. To better protect patient privacy, recent advancements have introduced source-free DA algorithms, where the learned model only has access to unlabeled target data during the adaptation process, bypassing the need for data sharing [84,85]. This approach allows for the development of models that can adapt to different domains without compromising patient data security. Integrating these source-free DA techniques into our FL framework will enable us to adapt the ML model to different hospitals without the requirement of sharing patient data (particularly all 18 HIPAA identifiers for protected health information), ultimately improving the overall accuracy of the model. By combining FL and DA techniques, we aim to address both the challenges of data decentralization and domain discrepancy, ensuring the development of accurate and robust ML models while safeguarding patient data privacy (Figure 2).

## 5. Conclusions

AI is increasingly being adopted in the field of medicine to derive actionable insights from extensive patient data and alleviate the workload of specialists. Shortcomings in current models utilizing nCLE for risk stratification of PCLs include the necessity for manual image pre-processing and the exclusion of pertinent data with significant predictive value. Our research group has previously developed nCLE-based CNN models for risk stratification of BD-IPMNs, which exhibited diagnostic accuracy surpassing that of standard-of-care guidelines, albeit using expert-edited endomicroscopy videos. We now present our strategy for enhancing the model’s accuracy and generalizability, which encompasses a video summarization algorithm, an integrated ML algorithm that harnesses a vast amount of data, and the validation of our predictive model within a multi-center cohort. While ML and AI require extensive imaging and datasets, the aspiration of this methodology review is to encourage researchers worldwide to collaborate in enhancing the management of patients with pancreatic cystic lesions.

## Figures and Tables

**Figure 1 biomimetics-08-00496-f001:**
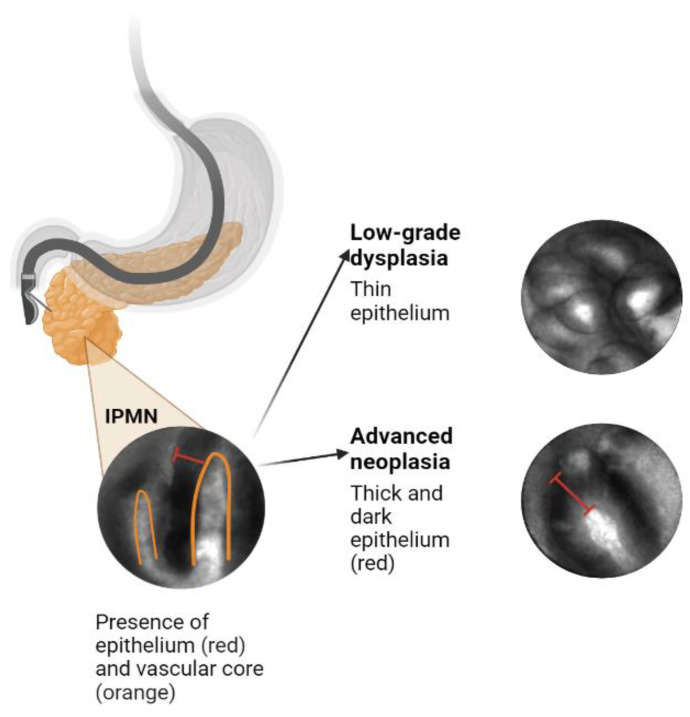
EUS-nCLE features of IPMNs.

**Figure 2 biomimetics-08-00496-f002:**
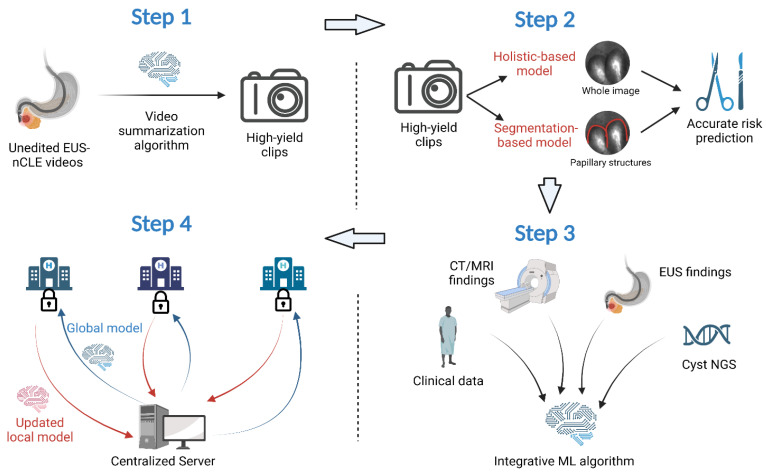
AI-guided EUS-nCLE image interpretation and PCL risk prediction.

**Table 1 biomimetics-08-00496-t001:** Limitations of existing AI models to diagnose pancreatic cystic lesions.

Study	Sample Size	Task	AUC	Shortcoming
Chakraborty et al., 2018 [37]	103	Predict high-risk IPMNs	0.77	Required manual outlining of pancreas images
Wei et al., 2019 [36]	260	Diagnose serous cystadenomas	0.767	Required image annotations by radiologist
Si et al., 2021 [34]	347	Diagnose pancreatic tumors using unedited CT images	0.871	Clinical and laboratory variables not included
Chu et al., 2022 [38]	214	Classify mucinous vs. non-mucinous cysts	0.940	Manual segmenting of CT images
Liang et al., 2022 [39]	193	Differentiate between cystic lesion types	0.973	Radiological features were extracted by radiologists
Schultz et al., 2023 [40]	43	Predict high-grade IPMN dysplasia	Accuracy 99.6%	Manual selection of high-yield single EUS images

IPMN: intraductal pancreatic mucinous neoplasm; CT: computed tomography; EUS: endoscopic ultrasound.

**Table 2 biomimetics-08-00496-t002:** Integrative algorithms for PCL risk stratification and PDAC diagnosis.

Study	Variables	Outcome	Performance (Individual)	Performance (Integrated)
Springer et al., 2015 [52]	Clinical + fluid molecular markers	PCL diagnosis	Sensitivity 76% (molecular)	Sensitivity 94%
Permuth et al., 2016 [60]	CT + fluid microRNA genomic data	Identify malignant IPMNs	AUC 0.77 (CT)	AUC 0.92
Springer et al., 2019 [58]	Fluid molecular data, imaging features, clinical data	Need for surgical resection of PCL	Accuracy 56% (standard of care)	Accuracy 69%
Kurita et al., 2019 [61]	Tumor markers, cyst location, cytology	Benign vs. malignant PCLs	Accuracy 71.8% (CEA), 85.9% (cytology)	Accuracy 92.9%
Blyuss et al., 2020 [62]	Urine biomarkers and CA19-9	Early stage PDAC detection	Sensitivity 81% (biomarkers)	Sensitivity 96%
Liang et al., 2022 [39]	Clinical + CT features	IPMN vs. MCN differentiation	AUC 0.90 (CT)	AUC 0.97
Hernandez-Barco et al., 2023 [63]	Demographic, clinical, imaging features	Low vs. high-grade dysplasia in IPMN	-	Accuracy 77.4%

IPMN: intraductal pancreatic mucinous neoplasm; CT: computed tomography; AUC: area under the curve; MCN: mucinous cystic neoplasm.

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
