# Peer review of "Improving Pancreatic Cyst Management: Artificial Intelligence-Powered Prediction of Advanced Neoplasms through Endoscopic Ultrasound-Guided Confocal Endomicroscopy"

_biomimetics, 2023, doi:10.3390/biomimetics8060496_

Round 1
Reviewer 1 Report
hank you for giving me a chance to review this article. This review includes relatively precursive modalities for diagnosing pancreatic cystic lesions. The authors should provide more detailed data of cited articles and make broader statistical analyses than current content (please ad more detailed tables). In addition, all cited articles did not diagnose IPMN; therefore, there is a descrepancy between the title and the content.
Reviewer 2 Report
The authors performed a review for primary articles describing original AI algorithms for diagnosing or predicting the malignant potential of pancreatic cystic lesions.
I read the study with interest. The study is interesting and well designed, although several major methodological problems were identified during review. These should be addressed and corrected. My concerns are as follows:
1. Title – It is unusual to use abbreviations in the title. Please revise.
2. The authors did not perform an optimal search of the literature. An optimal search in a review and meta-analysis requires searching at least 4 databases. (The authors searched only 2 databases). This should be revised and the results of the fourth database search should also be included. Please see the link below:
https://systematicreviewsjournal.biomedcentral.com/articles/10.1186/s13643-017-0644-y
3. Authors should include the exact date they conducted the literature search and the initials of the authors involved in the search.
4. The authors should include a flowchart of the study according to PRISMA guidelines.
5. Outcomes or even objectives of the study were not mentioned. Please state the aim of the study and clearly state the primary and secondary outcomes of the study.
6. Table 1 – Any abbreviations used in the table (e.g. IPMN, AUC, CT...) should be listed in a table legend.
7. The interpretation of the results is somewhat verbose and some parts should be moved to the Discussion. On the other hand, the discussion is sparse and should be expanded. I would advise replacing some parts (which are interpretation and not actual results) with discussion.
8. At the end of the discussion, the limitations of the studies included in this review should be listed.
9. After the discussion, add a separate chapter – conclusion - and end this review with some of the main findings.
Reviewer 3 Report
This is a very fine review on a promising solution of the clinical dilemma of incidental pancreatic cysts. Although there are existing algorithms to deal with this problem, their accuracy is not satisfying. This review points out which possibilities may be the future in predicting the risk of PCL.
The only item I propose is, that possible problems to reach an AI powered prediction as well as requirements for validation should be addressed
Reviewer 4 Report
1) The authors are using the full names along with their abbreviations again and again, please see pancreatic cystic lesions (PCLs), Artificial intelligence (AI), and machine learning (ML), among others. They are abbreviated in the abstract. Then, in the introduction section also.
2) I would recommend removing the abbreviations from the abstract.
3) Page 44 and 45, the author said "Conversely, patients with only low-grade dysplasia (LGD) can be managed through serial imaging monitoring". Provide a reference or online link for it.
4) Some references are cited number-wise and some references are cited year-wise. The authors should use a unified format.
5) Line 214, the authors said that "The accuracy of our existing EUS-nCLE model can be enhanced". I did not understand it. The authors talk about the existing model or the new proposed model. If the author talk about the existing models, then, provide a reference.
6) If the author talk about the new proposed model. Then, the authors should also discuss the disadvantages or limitations of the proposed model.
7) Some hints or guidelines about improving these limitations are welcomed.
Minor editing of English language required
Round 2
Reviewer 1 Report
The authors have revised their manuscript along with reviewers' suugestions.
Reviewer 2 Report
The authors significantly improved the quality of the manuscript. In my opinion manuscript is acceptable in its present form.
Reviewer 4 Report
The authors have addressed all concerns and implemented my comments. The paper is much better now. Therefore, I recommend its acceptance for publication.
Minor editing of English language required